# Brain Plasticity in Humans and Model Systems: Advances, Challenges, and Future Directions

**DOI:** 10.3390/ijms22179358

**Published:** 2021-08-28

**Authors:** Luca Bonfanti, Christine J. Charvet

**Affiliations:** 1Department of Veterinary Sciences, University of Turin, Largo Braccini 2, 10095 Grugliasco, TO, Italy; 2Neuroscience Institute Cavalieri Ottolenghi (NICO), Regione Gonzole 10, 10043 Orbassano, TO, Italy; 3Center for Neuroscience, Delaware State University, Dover, DE 19901, USA

**Keywords:** neurogenesis, immature neurons, translating time, mammals, doublecortin, PSA-NCAM

## Abstract

Plasticity, and in particular, neurogenesis, is a promising target to treat and prevent a wide variety of diseases (e.g., epilepsy, stroke, dementia). There are different types of plasticity, which vary with age, brain region, and species. These observations stress the importance of defining plasticity along temporal and spatial dimensions. We review recent studies focused on brain plasticity across the lifespan and in different species. One main theme to emerge from this work is that plasticity declines with age but that we have yet to map these different forms of plasticity across species. As part of this effort, we discuss our recent progress aimed to identify corresponding ages across species, and how this information can be used to map temporal variation in plasticity from model systems to humans.

## 1. Introduction

Many strategies used to alleviate disease or brain damage rely on plastic features of the nervous system. Stem cells have much potential for brain repair [1,2]; nevertheless, their occurrence and endogenous regenerative capacity have limits in mammals [3,4,5,6,7]. The nervous systems of some non-mammalian vertebrates (e.g., reptiles, axolotls, fish) have striking potential for regeneration [8,9,10] relative to mammals [3,4,7]. Yet, mammals possess multiple forms of plasticity; examples include neuron and glial production, synaptogenesis, and post-neurogenetic maturation [1,7,11,12,13,14,15,16]. These plastic features vary spatio–temporally and enable brain circuits to be sculpted based on experience [7,17,18,19,20]. A theme to emerge across studies is that plasticity (e.g., cell production, synaptogenesis) diminishes with age [21,22]. Given that species vary widely in their length of development and lifespan, it is critical that we consider variation in schedules of development and aging to accurately map temporal variation in plasticity from model systems to humans [23].

The environment influences plasticity, susceptibility to disease, as well as schedules of development and aging [18,19,24,25,26,27,28]. These features align with the notion that the environment impacts development at the anatomical, cellular, and molecular scales [17,28,29]. Most of these aspects, especially those based on synaptic plasticity, have been addressed for decades in the context of critical periods [13,30]. Yet, after a flourishing of studies on other types of neurodevelopmentally regulated structural changes (e.g., adult neurogenesis and, more recently, the non-dividing “immature” neurons), additional biological programs fall under the purview of “sensitive windows” (see [31,32] and below).

It is an open question as to which types of plasticity, and their time courses, are sufficiently conserved from model systems to humans to be relevant to human health. For instance, there has been an extensive debate as to whether hippocampal neurogenesis declines to non-existent levels in adulthood [33,34]. Some of these controversies stem from the fact that individuals used in these studies vary in age. Aligning ages across species permits mapping and, thus, comparing temporal declines in hippocampal neurogenesis across species [35]. One recent study revealed that hippocampal immaturity declines similarly with age across model systems and humans once variation in developmental schedules is accounted for. Translating ages across species addresses which biological processes vary together and which vary separately. Efforts such as these enhance translational potential from model systems to humans. We discuss past and recent progress on the translating time web resource originally developed by Dr. Clancy and her collaborators [36,37,38,39]. This online resource finds corresponding ages across species and can be used to assess how plasticity (e.g., neurogenesis, neuron immaturity, synaptic plasticity) has been modified between humans and model systems. 

## 2. Plasticity: What Is It and How Does It Vary?

There are lacunae in our understanding of plasticity, especially as to whether and how plasticity is an extension of “immaturity.” Nevertheless, some key insights are well established. First, most forms of plasticity are retentions of “embryonic” or “immature” features, some of which extend into postnatal ages [40,41,42,43,44,45,46]. Second, plasticity and immaturity progressively decrease with age [21,47]. Third, plastic changes vary by brain region, such that there is “differential” maturation across regions and cell populations (see below). Fourth, structural changes embrace a wide range of “types and scales,” with microscopic modifications affecting small portions of pre-existing cells (neurons, glia; e.g., synaptic plasticity) and more macroscopic changes varying at the level of cell numbers (adult neurogenesis, gliogenesis). Fifth, plastic changes vary remarkably among mammalian species according to their developmental schedules [32,35,36,37,48,49,50,51,52].

These distinctions appear semantic but distinguishing them builds a shared terminology. Whether adult neurogenesis or “plasticity” is intended to describe dendritic or synaptic changes of newly born neurons is not always clear [53]. The genesis of neurons followed by their post-neurogenetic maturation can be viewed as two distinct processes, or as one multistep process, involving cell division, cell specification, migration, differentiation, and integration through synapse formation [14,54,55]. Similarly, neurogenesis spans prenatal and postnatal ages extending into adulthood in select regions such that there is no clear distinction between neurogenesis in development and in adulthood [44,45,56]. As a consequence, confusion arises in reference to adult neurogenesis, immaturity, as well as plasticity. Immaturity is a temporary state of the cell, whereas plasticity is an action. These points raise the issue that plasticity may occur in the absence of immaturity (e.g., the retention of a cytoskeletal molecular machinery allowing structural modifications of the cells or the existence of a stem cell “niche”). The relationship or meaning of “maturation,” “plasticity,” and “senescence” are not always clear. Does “immaturity” enable plasticity, and delay senescence? Is a mature brain antecedent to senescence? Plasticity spans multiple stages of development, including neurogenesis, cell integration, as well as neuronal loss. It is timely to consider forms of plasticity over the course of development to unravel the cellular, molecular, and genetic substrates underlying neuron and brain maturation, and the biological significance of retaining immaturity in the CNS (see also [57] and below).

Brain plasticity is intrinsically linked to critical periods. These are “temporal windows” during which neural circuits can be modified by activity and experience, both of which are necessary for proper neural circuit assembly [30]. Studies of visual, auditory, and somatosensory maps have historically focused on synaptic plasticity during early postnatal development [13]. Adult neurogenesis, which was highlighted as an important biological process in the 1990s, spans critical periods in select regions, and fosters the retention of immature features (e.g., the olfactory system; see [31]), though the timelines of neurogenesis are characterized by remarkable species differences [52,58,59].

Plastic processes are highly heterogeneous (Figure 1, Figure 2B and Figure 3B). At least three main types of structural plasticity are recognized. First, synaptic plasticity, which refers to modification in contacts between neurons through the formation and elimination of synapses [11,60]. Such plasticity impacts contacts among pre-existing neurons and, thus, may be considered a form of immaturity persisting only at the very tip of the neuronal processes. These possibly occur within a neural network substantially assembled and “mature.” Second, adult neurogenesis is intended as the stem cell-driven formation of new neurons in the postnatal brain [1,14,15,61]. The extension of embryonic neurogenesis requires stem cell division maintained by a stem cell niche (an “immature” or “embryonic-like” environment [62]). This is important because neurogenesis in the adult brain requires both active stem cells within a conducive environment (niche). These are features present in very specific neurogenic sites [44,63]. Third, a later-discovered population of “immature” neurons (Ins) are generated prenatally and continue to express genes indicative of immaturity throughout adulthood in the cerebral cortex [12,64]; see below. This population of “non-newly generated, immature” neurons (nng-Ins) has been considered a novel form of “neurogenesis without division” [12,64,65,66,67,68]. The cortical nng-Ins might represent a novel form of plasticity by maintaining an undifferentiated (but regionally specified) population of young neurons in a region not endowed with stem cells and their niches. Neurons can delay immaturity “at an intermediate level”, by providing undifferentiated elements within a neuronal network otherwise already “mature” and in the absence of cell division [7]. Since these cells persist through the lifespan (though progressively decreasing in number [32,66]), they may be considered as a variant on the theme of critical periods remaining open for a long time.

Recent studies highlighted that the landscape is more complex, since in other brain areas, e.g., some subcortical regions, a mix of neurogenic and non-neurogenic plastic processes concomitantly exist, possibly varying by age [69,70,71]. In addition, several reports revealed remarkable phylogenetic variation in rates and types of plasticity across mammals [32,52,59,72], suggesting that some plastic processes underwent an evolutionary trade-off on the basis of brain size [32,52,73]. Synaptic plasticity, adult neurogenesis, and “immature” neurons show substantial differences in their occurrence, location, and extent across the lifespan (Figure 2A) [32,52,59,73,74]. Stem cell-driven adult neurogenesis is substantially decreased in large-brained species [47,52,72,75,76], including humans [33,58,77,78], with respect to its high rate/lifelong features in laboratory rodents (Figure 2). On the other hand, the populations of nng-INs immature neurons (almost absent in the rodent neocortex) are abundant in the expanded neocortex of large-brained mammals [32]. For some of these structural aspects, the progressive reduction in plastic features over the lifespan is species-specific [33,58,77,78]. These observations may impact the validity of using laboratory rodents as models to study humans [70]. According to a recent interpretation, the postnatal genesis of neurons in mammals should not be considered as a classic renewal/regenerative process (similar to what happens in stem cell systems of other body tissues or in the fish brain), but as a rather slow process to complete the development of certain neural circuits [7,17,20,79]. Such processes appear to be linked to specific behaviors and socio-ecological specializations, which widely vary among mammals [72,80]. 

Considering various forms of plasticity together may maximize the probability of identifying therapeutic and preventive interventions for neurological disorders and the maintenance of brain efficiency through aging. A complete mapping of brain maturation heterogeneity in different mammalian species is at present lacking. Systematically mapping these different forms of plasticity across species would permit relating these data to humans.

## 3. From Neuronal to Brain Maturation: Definitions and Heterogeneity

Brain maturation is frequently carried out with neuroimaging techniques in longitudinal studies [28,29]. These methods have been used to investigate morphological changes across the lifespan in health and in disease and across large-scale populations. Nevertheless, the diverse methods used across studies may contribute to inconsistencies across the literature [81]. Given these metrics focus on the macro- rather than the micro-scales, probing molecular, histological, immunohistochemical, and genetic levels can provide more complete insights into biological programs occurring across cell and subcellular scales. The heterogeneity in cellular constituents and plastic processes make the integration of macro- and micro- scales a challenge. Bridging scales of study at the genetic and neuroimaging scale will yield a more complete understanding of biological programs underlying plasticity [23]. 

In a recent review article [57], neuronal maturation at the cellular level has been addressed and conceptualized as “a research field that will have a strong impact on understanding the healthy and diseased nervous system.” The authors further state that “identifying the key mechanisms underlying neuronal maturation has the potential to reverse this process in adulthood, thereby facilitating regeneration.” In this case, they were referring to axonal regeneration [82], a process occurring in the mammalian CNS, which is very different from “brain regeneration”. Regenerative events are hardly present in mammals [4,5,7] (Figure 1), and are a feature of non-mammalian vertebrates that retain widespread stem cells through adulthood as well as the ability to re-activate developmental programs [9,83].

Until now, the issue of brain maturation related to age has been under-estimated, and often addressed from different angles by neurobiologists, neuroimagers, and neuropathologists. More interdisciplinary approaches are required. In particular, there is a need for improvements in defining “maturation”, as it pertains to neurons, and its relevance to “plasticity” in its different forms (Figure 1 and Figure 2). The cellular and molecular aspects of neuronal maturation have been extensively studied (though we surely need a deeper understanding; see [57]). On the topic of brain maturation, more work is needed, especially considering the emerging phylogenetic variation [32,52] (Figure 2). Furthermore, it is becoming increasingly clear that other than cell-intrinsic properties, a crucial role in allowing (or hampering) structural plasticity is played by the tissue environment in which the cells live, including the extracellular matrix and the perineuronal nets [84,85,86,87]. Accordingly, plasticity should be defined as a balance between the cell-intrinsic potential for structural changes (depending on neuronal maturation) and the limits or opportunities provided by the surrounding environment (depending on brain maturation). 

A key open question is: how is the process of maturation differentially regulated in specific cell populations or neural circuits? We know that structural changes occur at specific locations wherein a proper “immaturity” of the local brain environment can allow them, such as in the dentate gyrus of the hippocampus, in the subventricular zone, and in the olfactory bulb layers [84,85,86,88,89]. Yet, it is only in the hippocampus that the site of neuronal maturation and integration is proximal to the site of genesis. This is in contrast with the olfactory bulb, which is situated millimeters away from the periventricular stem cell niche, and in which migration occurs within astrocytic “glial tubes” [90,91]. In the case of cortical nng-INs, the situation is different (and substantially unexplored) since there is no recognizable “niche” or neural progenitors [7]. Thus, the question is: how are these neurons maintained in an immature state within no apparent immature microenvironment? Is there an involvement of perineuronal nets or surrounding astrocytic processes [65,92]? A recent report in which the polysialic acid polymer of the PSA-NCAM was enzymatically removed showed an increased maturation of the cortical nng-Ins in the mouse piriform cortex [93]. These data showcase the importance of the pericellular microenvironment in the maturational process.

In addition to heterogeneity across different “types of plasticity”, the occurrence and features of structural changes can vary extensively across mammals [32,47,52,59,72,73,80] (Figure 2), highlighting the complexity in translating these findings from model systems to humans. While synaptic plasticity is considered to span the CNS in a rather invariant way among species, adult neurogenesis and the presence of “immature” neurons show substantial differences in their occurrence, location, and amount across the lifespan of mammals [32,52,59] (Figure 2). Much work is yet to be conducted to track different forms of brain plasticity and immaturity over the lifespan and in different species. 

Finally, an important, emerging question is: can maturity be reversed in specific cell populations? Apparently yes [94], but it is not clear whether this fact might lead to positive outcomes, at least in mammals. In non-mammalian vertebrates, (e.g., fish), a process of de-differentiation can be activated after brain damage or inflammation, which leads to regenerative processes involving stem/progenitor cells and the reactivation of developmental programs [9,83]. In mammals, this does not happen in an organized manner, though reactive neurogenesis from endogenous neurogenic sites or reprogramming of parenchymal astrocytes that re-acquire stem cell properties can occur [95,96,97]. Nevertheless, these reactive changes usually do not lead to an effective tissue regeneration, mostly remaining as aberrant/abortive reactions likely due to an unfavorable tissue environment [7]. Finally, the outcome of reactive neurogenesis and cell reprogramming should not be confused with another process called “dematuration”, which is associated with the re-expression of molecules associated with immaturity (e.g., DCX and calretinin), and which has been described to occur in old individuals as a consequence of inflammation, neuronal hyperexcitation, and drug administration [98]. Dematuration is a process in which “mature neurons dedifferentiate to a pseudo-immature status and re-express the molecular markers of neural progenitor cells and immature neurons” [98,99], and is different from de-differentiation. In summary, the complex, heterogeneous issue of brain maturation, as a prerequisite for plasticity, is important in normal CNS postnatal development, influenced by the environment [28], and potentially fundamental in preventing or curing a wide range of neurological disorders. 

## 4. Maturation in the Mammalian Brain According to Location, Age, and Species: Examples and Implications for Human Brain Plasticity

Postnatal/adult plastic processes span the genesis and assembly of the neuronal population/circuit (e.g., the postnatal genesis of the cerebellar granule cells) to the modifications at the tip of neuronal processes (e.g., synaptic plasticity). These occur in a substantially mature nervous tissue. Hence, at a certain time point, neural circuits must be necessarily endowed with different gradients of maturation. Such a “differential maturation” occurs in different brain regions and neuronal populations, and shows remarkable variation among mammalian species (even among individuals), leading to a highly heterogeneous landscape (Figure 2). Due to the complexity of this topic, a full description is still lacking, but we include a few examples.

(i) *Neurogenesis occurs postnatally (postnatal genesis and formation of neural circuits).* The production of cerebellar granule cells extends beyond birth, through the transient existence of a subpial germinative layer (the external germinal—or granule—layer) and, subsequently, ceases sharply after the exhaustion of progenitor cells [100]. Apart from their prenatal origin from progenitors located in the anlagen of the rhombic lip, all cerebellar granule cells are produced during the early postnatal period (about 3 weeks in mouse and 1 year in humans). The external granule layer expands from a thin subpial layer to six-to-eight-cell layers deep. Following clonal expansion, the granule cell precursors exit the cell cycle and migrate radially along the radially oriented Bergmann glia fibers. Post-migratory granule cells settle in the nascent granular layer, where they extend dendrites and form synapses with mossy fiber afferent axons [100,101] (Figure 2B). The genesis of cerebellar granular neurons, which outnumber cortical neurons’ numbers, display a wide range of plastic processes that are typical of neural development. Those include cell proliferation, specification, long-distance migration, differentiation, maturation, and the establishment of functional neural circuits. We have yet to translate the timeline of biological programs within the cerebellum across species to identify which developmental processes occur for an unusually long time in select taxonomic groups. 

(ii) *Neurogenesis extends postnatally (refinement of neural circuits by the addition of new elements).* Postnatal neurogenesis in two canonical neurogenic sites, the olfactory bulb and the hippocampus, has been extensively studied [14,15,44,102]. A recurrent bias or misunderstanding has been present since the discovery of adult neurogenesis. This biological process, first considered as a “regenerative” event similar to other stem cell systems in the body, is now recognized as a possibility of a progressive addition of new neurons in neural circuits that are already formed (and then are not continuously/completely renewed), in order to “sculpt” them on the basis of the animal experience, especially during youth [17,19,20,103]. Dissimilar to the correspondent processes in non-mammalian vertebrates, this prevalent “neurodevelopmental feature” of mammalian neurogenesis is accompanied by a scarce capacity for brain repair and regeneration [7].

Different roles and modes of integration have been observed for olfactory and dentate gyrus neurogenesis (respectively, maintenance and reorganization of some olfactory bulb neuronal populations versus the modulation and refinement of existing neuronal circuits [104]). Indeed, most of the old neurons are replaced by new neurons in the deep regions of the olfactory bulb, and half of the population is replaced in the superficial regions in mice. On the other hand, most neurons are maintained in the dentate gyrus where these new hippocampal neurons contribute to an increase the whole granule cell number during adulthood [104]. As discussed in a recent review article [31], the olfactory system neurogenesis shares little in common with classic models of critical periods, including ocular dominance, auditory maps, or barrel cortex plasticity. Present evidence argues that “it is unique among sensory systems” in displaying the potential for persistent circuit plasticity rather than the experience-induced functional rigidity and anatomical consolidation that characterizes classical models of a critical period [31]. Thus, olfactory bulb neurogenesis displays a singular form of neoteny, delaying immaturity or slowing maturation.

Both neurogenic processes can continue during adulthood in mammals, yet they present two important aspects. First, their rate remarkably decreases with age in all species considered, including laboratory rodents [22,35]. Second, they have been shown to be limited to early postnatal–young stages in large-brained mammals [35,52,58,59,76,78]. 

The problem of determining the mere existence of hippocampal neurogenesis in humans illustrates that translational tools are needed to extrapolate information from model systems to humans [35]. A previous study considered the relative number of hippocampal DCX+ cells in different species and at different ages [35]. The relative number of DCX+ neurons decreases with age, accompanied by a decrease in cell division, which reflects an actual decrease in adult neurogenesis. Of interest, similar patterns of decline in the relative number of DCX+ cells across species were observed once ages of model systems were aligned to human age. According to these data, the relative number of newly generated cells is expected to decline to hard-to-detect levels towards childhood. These data are consistent with the prediction determined by Sorrells et al. [33,78] who report that the relative number of newly born neurons decline to hard-to-detect levels during childhood in humans. It is clear from studies across humans and model systems that some immature neurons (DCX+ neurons) might persist in adulthood because of the availability of markers to capture such processes [16,99,105]; see below. Olfactory and hippocampal adult neurogenesis occur with different rates and time courses in humans. It would be of interest to investigate how neurogenesis vary more broadly across species once variations in developmental schedules are accounted for. 

(iii) *Non-newly generated, cortical “immature” neurons (nng-INs; neurogenesis without division).* These neurons are generated prenatally and retain the expression profiles of markers indicative of immaturity (see above). These neurons can progressively mature through the lifespan and integrate as functional neurons in the layer II of the cerebral cortex, namely, a brain region that is not endowed with stem cell-driven neurogenesis [65,66,67,93]. In this case, a whole cell retaining immature features is surrounded by a neuropil composed of mature neurons. It is thought that the cortical nng-INs possess few or no synapses and are isolated from the surrounding functional circuits (likely by a sheet of astrocytic lamellae [65]). The mechanism(s) allowing these neurons to stop differentiating at a certain stage of maturation are still completely obscure, and it is intriguing that they do not undergo cell death in the absence of connections with other neurons. All these aspects are worthwhile of further investigation since nng-INs might represent an evolutionary twist allowing the cerebral cortex to circumvent its lack of stem cell-driven neurogenesis. Accordingly, a remarkable phylogenetic variation characterizes the cortical nng-INs of mammals, with far larger amount (cell density) and regional extension (from paleocortex to neocortex) when shifting from small-brained to large-brained species [32]). 

(iv) *Refinement of neural circuits by modification of pre-existing elements.* Several types of structural changes fall into this category (e.g., synaptic plasticity, pruning, neuro-glial plasticity). Due to a large literature covering these aspects, we will not address these processes here. Rather, we underline the fact that, despite undoubted improvements concerning technological advances allowing the establishment and maintenance of neural circuit architecture [106,107], it is very difficult to quantify such processes among different species, brain regions, and ages. Another important chapter of postnatal brain maturation is linked to myelination, a process which contributes to critical periods and crystallized circuits [108,109]. The division of oligodendrocyte progenitor cells plays a role in myelination, which is a widespread phenomenon across the CNS and is more common than neuron production during postnatal development [79]. Additionally, these plastic processes show remarkable interspecies differences, which make it difficult to reconcile data obtained in mice and humans [6,110]. Hence, we have limits to our capability to define (and measure) the degrees of CNS tissue immaturity due to an intermix of different cellular and molecular features leading to plastic changes, which can vary among individuals as a consequence of their life experiences. 

Overall, these examples illustrate how very different processes and cell-environment interactions can be observed. For instance, cells producing the cerebellar granule cell layer act postnatally in an “embryonic-like” environment (see [101] for the molecular machinery involved), which is different from the modified environment characterizing the stem cell niche/neurogenic sites in adult neurogenesis [45], and completely different from the mature neural circuits surrounding the cortical nng-INs [12,32,65]. These wide-range situations depict the heterogeneity of brain structural plasticity across regions and ages. Of importance, new “subtle” forms of structural changes have been (and are) continuously demonstrated beside the more striking, and deeply studied, neurogenic processes. Recent reports highlight the importance of these new, non-canonical, forms of plasticity for large-brained mammals, including humans [7,12,16,32,59,64,99,111]. 

The importance of various forms of plasticity extending into adulthood is evident from multiple levels. Progressive maturation provides a window of opportunity for structural changes (plasticity) and enables cell populations to be shaped by life experience, and to adapt to different situations. The exclusive reliance on laboratory rodents as an experimental model to understand human biology is limiting. In this context, a novel approach for studying brain maturation across species (the importance of addressing this issue in primates) focuses on extracting time points across biological programs throughout ages to find corresponding ages across species. 

### Role of Brain Plasticity and Immaturity in Psychiatric and Neurological Disorders

It is well known that alterations in brain maturation and plasticity are linked to the emergence of neurological disorders [112,113,114,115,116,117]. In particular, schizophrenia and psychosis spectrum disorders are associated with an impacted timeline of brain maturation [29,118]. Though the precise mechanisms associated with the emergence of neurological disorders remain elusive, research in neuroimaging have started to shed some light on the problem. For instance, we know that the mis-wiring of prefrontal areas, resulting from genetic or environmental cues, act during brain development, and contribute to cognitive impairment in psychiatric disorders [112,114]. In schizophrenia, the dysfunction of local circuitry within the prefrontal cortex and its long-range connectivity is linked to aberrant maturation and connectivity [115]. The high adaptability of the adolescent brain might make it particularly vulnerable to an abnormal formation and refinement of connections [119]. Some psychiatric disorders, such as schizophrenia, anxiety, and depression, show an onset of symptoms towards the end of this maturational period [120].

Plastic changes, induced by lifestyle in the young and immature brain, are useful to cope with normal and pathological age-related cognitive decline [121,122], likely, through the establishment and implementation of the so-called “brain reserve” [24,27,103,123]. It is now clear that beneficial lifestyle choices may promote changes that enhance cognition, such as angiogenesis, synaptogenesis, and neurogenesis [124]. According to a recent interpretation, critical periods, in which experience instructs neural networks to develop into a configuration that cannot be replaced by alternative connectivity patterns, are different from sensitive periods. Experience leads to many possible network configurations or connectivity patterns that can compensate for each other and are subjected to remodeling [125]. Advances regarding the use of developmental manipulations in the immature brain of animal models are the basic preclinical systems that will allow the future translatability of timely interventions into clinical applications for neurodevelopmental disorders such as intellectual disability, autism spectrum disorders, and schizophrenia. Nevertheless, besides the involvement of neurotrophins, semaphorins, and GABA deficits [115,118,120,126], little is known about the molecular substrates regulating the relationship between plasticity, maturation, and neurological disorders. Since different sensitivity in plastic responses can be found at different, critical ages (postnatal stages, adolescence, youth, aging), it is not easy to translate results obtained in animal models (especially short-living laboratory rodents) to the lifespan and lifestyle of long-lived species such as humans. In summary, the evident involvement of plasticity and brain maturation across a wide range of psychiatric and neurological disorders (both in their pathogenesis and in the perspective of their prevention/treatment) must find more solid neurobiological bases, as well as further knowledge in the translating time science to enhance interpretation of findings from animal model-based studies [36,38].

## 5. Molecules of Immaturity: Their Limits and Use/Misuse as Markers

Most of the current knowledge concerning CNS “immaturity/maturity” relies on the immunocytochemical detection of markers. The main molecules used with this purpose have been reviewed [12,54,55,127,128,129]. Here, the aim is to underline their limits, since most of these markers are insufficient to characterize the nuances of immaturity, for at least two reasons. First, they are not always specific for the precise identification of maturational stages. Second, their expression can vary remarkably in different systems, ages, and species. As an example, we will focus on two of the most used markers: Doublecortin and PSA-NCAM (Figure 3).

Doublecortin (DCX) is a microtubule-associated protein expressed by migrating neuroblasts in both developing and adult mammals [130,131,132]. It plays a crucial role for microtubule stabilization, nuclear translocation during neuronal migration [133], and growth cone dynamics [134]. In humans, it is essential for normal brain development, and mutations cause X-linked lissencephaly with the characteristic defect in cortical layering [135]. This protein is expressed by several cell populations varying in morphology and across immature cell populations. DCX expression begins in dividing neuronal precursor cells and is then downregulated to undetectable levels when neurons begin differentiating [127]. Through the years, these characteristics led to the misunderstanding that DCX could be a specific marker for adult neurogenesis. Yet, we now know that it can be found in widely different CNS cell populations, including non-newly generated neurons [12,16]; see below.

PSA-NCAM (polysialylated form of N-CAM; “embryonic” N-CAM), a member of the immunoglobulin superfamily of adhesion molecules, is an anti-adhesive form of N-CAM [136,137]. Polysialic acid (PSA) is a large carbohydrate added post-translationally to the extracellular domain of the transmembrane protein N-CAM; thus, leading to a reduction in the interaction between cell expressing PSA-NCAM and the surrounding neuropil [138]. In the developing nervous system, it promotes dynamic cell interactions, such as those responsible for cell migration, axonal growth, terminal sprouting, and target innervation. In the adult nervous system, its expression is restricted to regions displaying different forms of plasticity and neurogenesis [127,137]. Similar to DCX, PSA-NCAM can be found both in newly generated neurons and in non-dividing “immature” neurons [12,16,32,64] (Figure 3).

DCX and PSA-NCAM are often co-expressed in newly born/young neurons and for this reason, in the past, were commonly considered associated with striking aspects of structural plasticity, such as adult neurogenesis and cell migration [127,128,132]. Nevertheless, it is now clear that they are also expressed by resident neurons (non-newly generated, non-migrating cells) that are generated during embryonic development (e.g., the nng-INs in layer II of the cerebral cortex [12,16,65,69,93,139,140,141]). In addition, DCX and PSA-NCAM expression has been described in similar cell populations located in subcortical regions (both in white and grey matter), and it is far from clear whether they are generated postnatally or not [69,70,71,142,143,144]. Hence, it remains an open question as to whether DCX is transiently expressed in immature neurons for similar lengths of time in humans as in rodent and how such transient expression changes with age and in different species [16,59,71,99,145]. Finally, PSA-NCAM is also expressed in many regions of the adult brain in association with neuronal-glial and synaptic plasticity (e.g., in some hypothalamic nuclei [138,146]).

While these markers can be used to detect “states of immaturity”, they cannot be selectively linked to specific biological processes (see Figure 3B). The use of multiple markers overcomes this issue (e.g., co-expression of DCX/PSA-NCAM with cell proliferation markers to identify or exclude neurogenesis) and provides internal positive controls [33,59]. The landscape of maturity/immaturity in the CNS is more complex due to interspecies differences. Considering diverse mammalian species from rodents to large-brained mammals raises the issue that cell populations across brain regions vary remarkably in the expression of the abovementioned markers. This is the case of the cortical nng-INs, which are abundantly present in the neocortex of sheep, cats, and chimpanzees, but are restricted to the piriform and entorhinal cortex in laboratory rodents [32]. 

## 6. Building on Translating Time to Align Ages across Species

Aligning temporal variation in plasticity in different species informs which biological programs are conserved, and which occur for an unusually long or short time in humans. One application to aligning ages is its ability to map temporal patterns of plasticity from model systems to humans. We developed norming procedures to compare brain development and aging across species, and we illustrate how we can harness information from model systems to predict temporal patterns of plasticity in humans. 

The need to translate findings across model systems and humans led to the development of an online resource called the translating time project, which finds corresponding ages across the lifespan of humans and model systems. The “translating time” model (www.translatingtime.org, accessed on 28 July 2021) finds corresponding ages during fetal stages of development across 18 mammalian species and humans, which extends up to 2 years of age in humans and their corresponding ages in other species. The resource relies on time points captured from abrupt changes in developmental programs across species to find corresponding ages mostly during fetal stages of development across humans and model systems. Focusing on abrupt transformations, they are especially well suited to finding corresponding time points at prenatal stages of development, but abrupt transformations become increasingly sparse with age. We, therefore, developed new methods to find time points at postnatal ages in humans and model systems to harness abrupt and gradual changes in the timing of biological programs to extend the translating time model to span postnatal ages (Figure 4 and Figure 5).

We now integrate time points from abrupt and gradual changes that span pre- and postnatal ages. These include time points extracted from temporal variation in transcription as well as structural variation. We successfully relied on abrupt changes as well as gradual changes in biological programs to find time points at postnatal ages across humans, chimpanzees, macaques, and mice [23,37,38,39]. We applied this approach to find equivalent postnatal ages in humans, macaques, chimpanzees, and mice. We implemented various statistical procedures to find corresponding ages across species. We identified corresponding time points from humans to mice up to 30 years of age in humans, which corresponds to 6 months of age in mice. For example, this work showed that a mouse on postnatal day 4 corresponded to a human at gestational week 21. A 6-month-old mouse corresponded to a 30-year-old human (Figure 4). We also aligned ages at pre- and postnatal ages across the lifespan of humans and chimpanzees to identify time points during pre- and postnatal ages up to 47 years in chimpanzees, which corresponds to 55 years of age in humans [38]. This recent study revealed a surprising degree in the conservation in the timeline of development and aging across chimpanzees and humans. Studies such as these are essential because they provide a baseline with which to compare the time course of biological programs across species to predict temporal patterns of plasticity in humans and potentially assess which biological processes span a usually short or long period of time in humans relative to other species. The ability to harness information from model systems to humans is particularly important because some information can be rather sparse in humans [33,34].

A baseline to compare the biology of humans and chimpanzees makes us better equipped to assess which biological programs are conserved and which have evolved in the human lineage. It also enables predicting temporal variation in biological programs in humans from model systems. We previously predicted a temporal pattern of variation in hippocampal neurogenesis in mice to predict the temporal pattern in humans [35]. We considered temporal variation in the relative number of DCX+ cells in the hippocampus at different postnatal ages in mice and non-human primates to predict the temporal pattern of hippocampal neurogenesis in humans (Figure 5). If the temporal pattern of hippocampal neurogenesis is conserved across rodents to humans, hippocampal neurogenesis should decline sharply during childhood to hard-to-detect levels. These findings align with reports by Sorrells and colleagues [33,78] (Figure 5). We have yet to align other forms of brain plasticity from model systems to humans. It is unclear whether transient expression in DCX spans similar timescales in humans and model systems and whether the relative number of DCX positive cells captures neuron production or neuron immaturity. Nevertheless, the translating time resource provides a venue with which to translate those findings across species and investigate these various forms of plasticity across species. 

New tools are needed to identify different types of plastic changes and to map CNS plasticity and maturation through time and across species. There are clear heterochronies in the timeline of select biological programs across species [36,51,147,148,149,150]. For example, neurogenesis in the presumptive primary visual cortex is extended in primates compared with rodents after controlling for variation in schedules of development across species [36,51,149,151,152]. The consequence of extending the duration of cortical neurogenesis is that primates possess disproportionately more neurons in the primary visual cortex relative to many other mammals [152,153]. The increase in neuron numbers in primates is driven by upper layer neurons, which are generated late in development [152]. Of interest is the finding that the extension in cortical neurogenesis does not entail an extension in the duration post-neurogenetic maturation as assessed by temporal profiles in the expression of neuronal markers [50]. That is, modifications in the timeline of neuron production are dissociable from post-neurogenetic maturation. This is one example in which the timeline of neuron production and post-neurogenetic maturation are dissociable. Similar dissociable events may exist in other brain regions [148,150], though we have yet to uncover the various ways in which the timelines of biological programs evolve independently across mammalian species.

## 7. Conclusions

Brain maturation is pivotal for cognitive development and the maintenance of efficiency and adaptation through plasticity. Although adult neurogenesis is often the focus of study, there are other manifestations of neuroplasticity, including nng-INs (neurogenesis without division), alterations in morphology (e.g., increased dendritic spines, modified dendritic branching) and neurophysiological functions (e.g., long-term potentiation, neural networks [7,154]). Such neural modifications provide opportunities for targeted interventions to maintain healthy brain function throughout life [154,155]. Recent research indicates that such goals are feasible, but the complexity of their cellular substrates is high. In addition to awareness of the heterogeneity of plastic processes and forms of immaturity, relatively recent studies started to highlight remarkable species differences in various forms of plasticity, which challenges the value of rodent models to understand human biology. The present work illustrated the importance of using a comparative approach to harness information across species to understand plasticity in humans. Yet, there is so much we have yet to investigate. It is clear that developing resources to integrate findings across model systems to humans is broadly applicable to enhance our understanding of human biology and health. Accordingly, comparative neuroscience applies broadly to enhance our understanding of basic biological processes in the human brain.

## Figures and Tables

**Figure 1 ijms-22-09358-f001:**
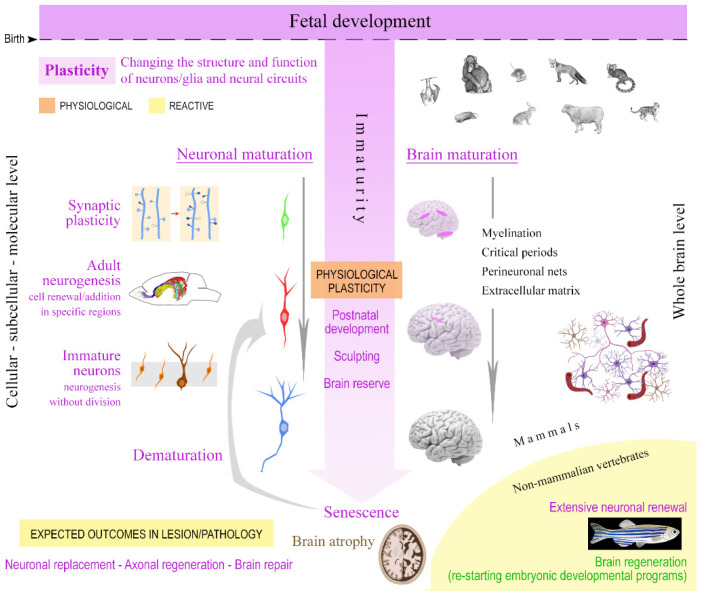
There are different types of structural plasticity which are linked to neuronal and brain immaturity. On the left are listed the main types of plasticity considered at a cellular/subcellular level (from the small modifications of synaptic contacts between pre-existing neurons to different types of “addition” of neuronal elements). On the right, the same processes are viewed at the whole brain level across the lifespan, determining differential maturation of various neural systems/circuits (brain regions) with remarkable differences depending on the animal species. Most of the naturally occurring plasticity plays a “physiological” role in brain postnatal development/sculpting and adaptation (center). Bottom, possible outcomes after lesion/disease (reactive plasticity) are limited in mammals, the main role of brain immaturity/plasticity being related to the postnatal brain development. A substantial difference in brain regeneration capability exists between mammals and non-mammalian vertebrates.

**Figure 2 ijms-22-09358-f002:**
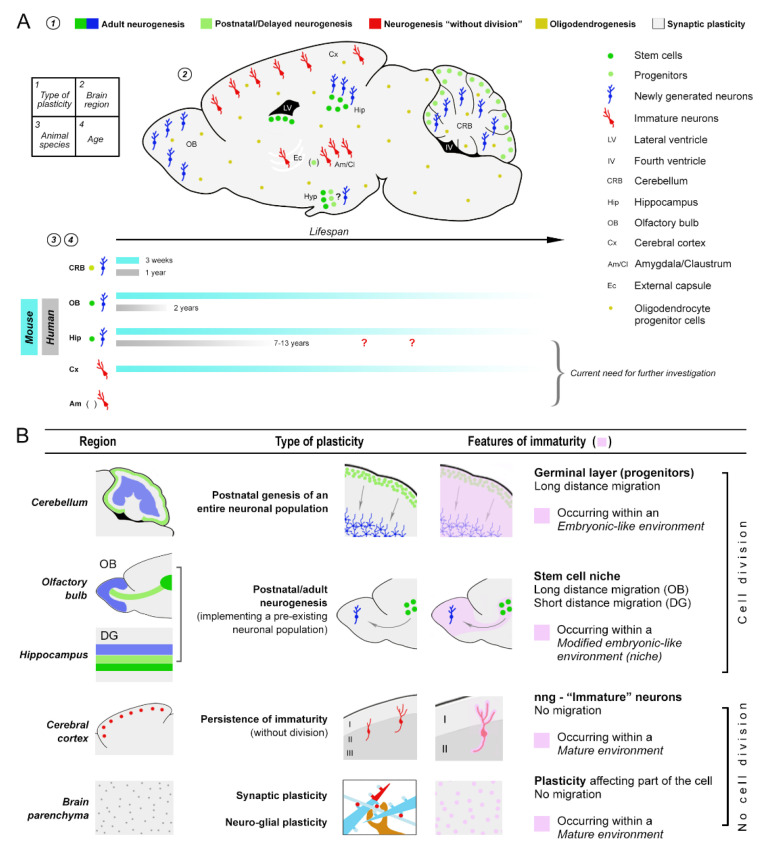
Immaturity persisting prior to the full maturation of the nervous tissue can grant plasticity, yet the processes involved occur differently depending on four main variables. Plasticity varies by species, brain region, and age. (**A**) Sagittal view of the mouse brain is used to represent mammalian brain regions (e.g., the immature neurons are not present in the mouse neocortex but they can be found in many large-brained species; see text) and to summarize the extreme heterogeneity of plasticity/immaturity in mammals. Due to a lack of comparative (and comparable) quantitative studies across widely different species, the current knowledge is far from being complete and merits further investigation in large-brained mammals in order to integrate data obtained in laboratory rodents. (**B**) A brief survey of the main biological (plastic) processes involved in brain plasticity and maturation, adding detail to scheme (A), highlight the heterogeneity of variables to be taken into account by comparing animal models.

**Figure 3 ijms-22-09358-f003:**
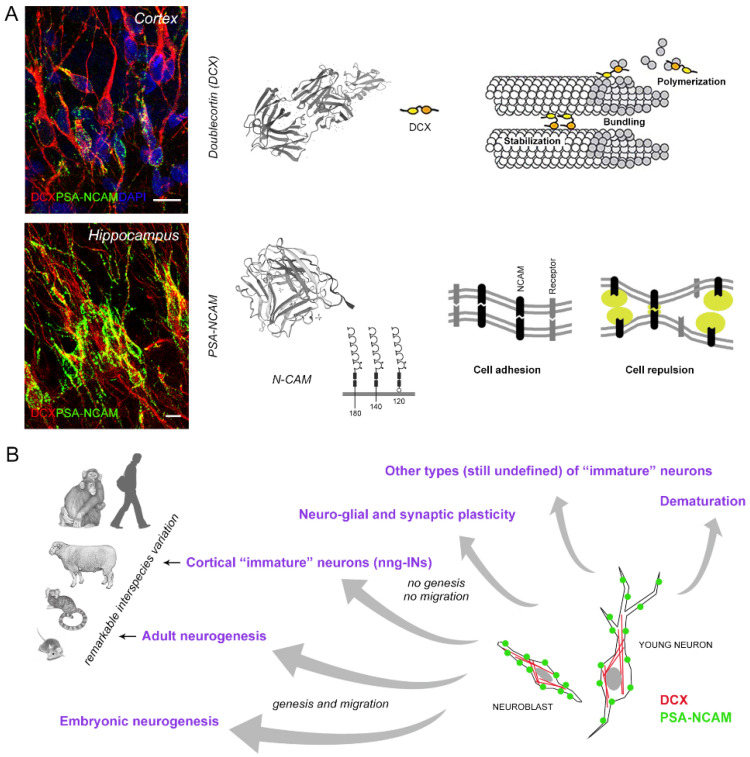
(**A**) Two molecules defining neuronal immaturity and most used as immunocytochemical markers for studying different forms of plasticity, including neurogenesis, adult neurogenesis and “immature” neuron detection. Doublecortin is a cytoskeletal protein involved in multiple structural changes of the cells; PSA-NCAM is an anti-adhesive, membrane-bound protein endowed with polymers of sialic acid on the extracellular domain. Scale bars, 10 µm. (**B**) DCX and PSA-NCAM are frequently co-expressed in cell populations associated with biological processes spanning a very wide spectrum, from neurogenesis to senescence. For this reason, these markers (if used alone, for instance, not in association with other –, e.g., cell proliferation markers) cannot specifically identify one of these processes. In addition, though sharing the same markers, some plastic cell populations remarkably vary in their amount and distribution depending on animal species among mammals.

**Figure 4 ijms-22-09358-f004:**
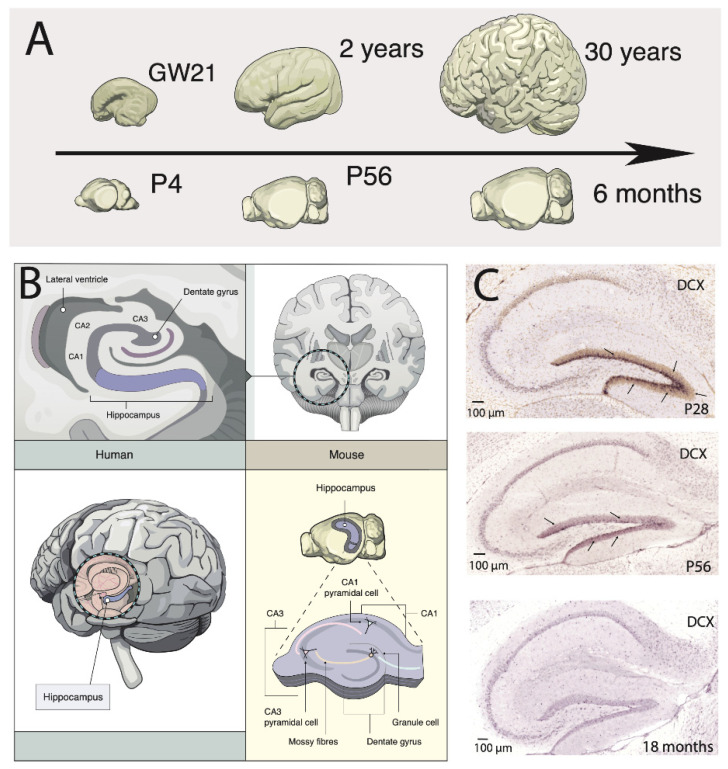
(**A**) We identified corresponding ages between humans (top panel) and in mice from abrupt transformations in behavior, anatomy, and transcription [37,51]. According to these data, a 2-year-old human corresponds to a mouse on postnatal day 56. A 30-year-old human is equivalent to a 6-month-old mouse. Aligning ages enables identifying whether the gradual decline in DCX expression shares a similar temporal pattern in humans and in mice. According to these data, DCX is difficult to detect after post-natal day 56, which corresponds to 2 years of age in humans. (**B**) Schematics illustrate the location and spatial organization of the human and murine hippocampus. Hippocampal neurogenesis extends postnatally in the dentate gyrus. (**C**) Images of doublecortin (DXC) mRNA expression, used as a marker of immature neurons, show that DCX expressions (arrows) in the dentate gyrus gradually decline with age in mice. Expression levels are relatively high at postnatal day (PD) 28 but steadily decrease from postnatal day 28 to 18 months of age. By 18 months of age, DCX expression is difficult to detect in the murine dentate gyrus. ISH images are from the used in situ hybridization Allen Brain Institute (http://mouse.brain-map.org, accessed on 28 July 2021).

**Figure 5 ijms-22-09358-f005:**
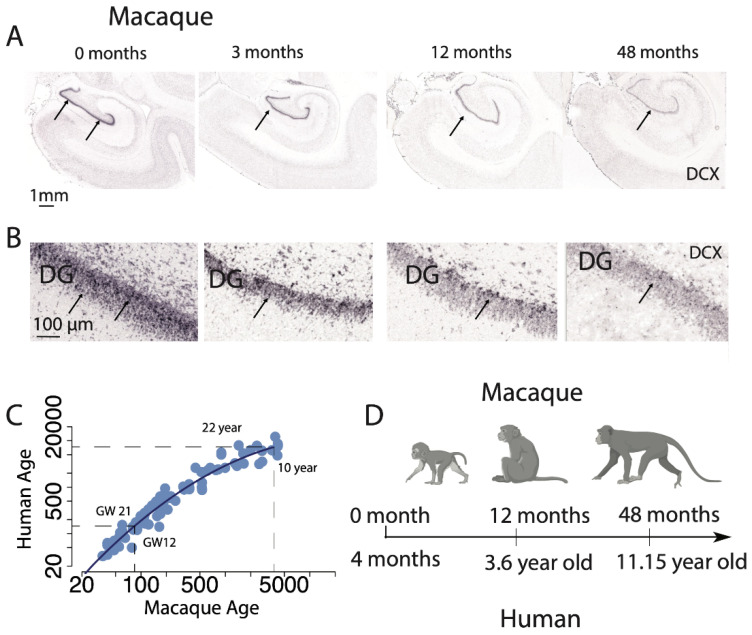
Low (**A**) and high-resolution (**B**) of DCX mRNA expression (arrows) in the macaque hippocampus from 0 months to 48 months after birth. These data show that DCX mRNA expression steadily decreases with age in the dentate gyrus (DG) to relatively low levels at 48 months of age in macaques. (**C**) We extracted time points from temporal variation in transcription, structure, and behavior in humans and macaques to find corresponding ages across these species [39]. According to these data, a macaque at gestational week (GW) 12 is equivalent to a human at GW 21 and, a 10-year-old macaque equates to a 22-year-old human. The identification of corresponding time points permits relating the temporal decline in DCX in across species. (**D**) For example, DCX expression in the DG was rather high between 0 and 3 months of age in macaques, which corresponds to 4 months after birth in humans. By 48 months of age, DCX mRNA sharply declined, which corresponds to 11 years of age in humans. Collectively, observations from mice and macaques pointed to a rapid decline in DCX mRNA during childhood in humans. ISH images are from in situ hybridization data made available by the Allen Brain Institute (http://www.blueprintnhpatlas.org; http://mouse.brain-map.org, accessed on 28 July 2021). Some panels are from BioRender.com, accessed on 28 July 2021.

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
