# Peer review of "Brain Plasticity in Humans and Model Systems: Advances, Challenges, and Future Directions"

_ijms, 2021, doi:10.3390/ijms22179358_

Round 1

Reviewer 1 Report

In the present review the Authors discussed and evaluated the most recent studies focused on brain plasticity across the lifespan and in different species. They suggested that that plasticity might decline with age but we have yet to map the different forms of plasticity across species. As part of this review, they also discussed the recent progress aimed to identify corresponding ages across species, and how this information can be used to map temporal variation in plasticity from model systems to humans.

Overall, I found the present review timely, well written, well organized and scientifically sound: enjoyed reading it! I have only some minor suggestions aimed to improve the high quality of the paper and these are explained below:

1) As brain maturation is pivotal for cognitive development and maintenance of efficiency and adaptation through plasticity, I wpuld suggest the Authors to add a brief paragraph on what it happens when plasticity is disrupted as it happens in psychiatric disorders as schizophrenia (see dois 10.3390/brainsci11020275 and 10.1093/schbul/sbt153).

2) A brief paragraph on how literature searches were conducted and relevant paper obtained and included  would be useful to the reader.

3) I guess what is the role of neurotrophins in brain plasticity and its decline with age.

Author Response

In the present review the Authors discussed and evaluated the most recent studies focused on brain plasticity across the lifespan and in different species. They suggested that that plasticity might decline with age but we have yet to map the different forms of plasticity across species. As part of this review, they also discussed the recent progress aimed to identify corresponding ages across species, and how this information can be used to map temporal variation in plasticity from model systems to humans.
Overall, I found the present review timely, well written, well organized and scientifically sound: enjoyed reading it!
- We thank Reviewer 2 for his/her positive recommendation.
I have only some minor suggestions aimed to improve the high quality of the paper and these are explained below:
1) As brain maturation is pivotal for cognitive development and maintenance of efficiency and adaptation through plasticity, I wpuld suggest the Authors to add a brief paragraph on what it happens when plasticity is disrupted as it happens in psychiatric disorders as schizophrenia (see dois 10.3390/brainsci11020275 and 10.1093/schbul/sbt153).
- We thank Reviewer 2 for this suggestion. We agree that this is a topic that is highly relevant to the present manuscript. In line with reviewer 2’s recommendations, we had also originally planned to discuss this topic. Yet, we chose not to address it in the first version of the manuscript because it is such a vast field. It is clear that the topic of neurogenesis is germane to disorders and should be included in some capacity. We summarized the topic in a paragraph at the end of section 4 (please see below). We also underlined the need for further investigation.
2) A brief paragraph on how literature searches were conducted and relevant paper obtained and included would be useful to the reader.
- The relevant literature was found on the basis of our experiences in the field, as well as on PubMed searches. We did not perform a systematic analysis of the literature or perform meta-analyses. This is a very broad field with roughly 11,000 papers on the topic of adult neurogenesis. Our review article is not intended to be a systematic analysis of this broad research area. Instead, it is a critical analysis of specific studies. We, therefore, reason it is not necessary to include
information about how literature searches were conducted since we were not systematic in our literature search.
3) I guess what is the role of neurotrophins in brain plasticity and its decline with age.
- Of course, neurotrophins are involved in brain maturation and plasticity and likely diminish with age. We do note that results concerning their occurrence, modulation, and relevance to neurological disorders and how these parameters vary with age have been controversial. We briefly addressed this issue in the new paragraph. Below, we paste our paragraph as well as the relevant references.
“4.1. Role of brain plasticity and immaturity in psychiatric and neurological disorders
It is well known that alterations in brain maturation, plasticity are linked to the emergence of neurological disorders (Selemon and Zecevic, 2015; Morton et al., 2017; De Berardis et al., 2021; Gao et al., 2021; Shor and Bianchi, 2021). In particular, schizophrenia and psychosis spectrum disorders are associated with impacted timeline of brain maturation (Fatemi and Folsom, 2009; Tamnes, 2017). Though the precise mechanisms associated with the emergence of neurological disorders remain elusive, research in neuroimaging have started to shed some light on the problem. For instance, we know that mis-wiring of prefrontal areas, resulting from genetic or environmental cues act during brain development, and contribute to cognitive impairment in psychiatric disorders (Selemon and Zecevic, 2015; De Berardis et al., 2021). In schizophrenia, dysfunction of local circuitry within the prefrontal cortex and its long-range connectivity is linked to aberrant maturation and connectivity (Gao et al., 2021). The high adaptability of the adolescent brain might make it particularly vulnerable to abnormal formation and refinement of connections (Chini and Hanganu-Opatz, 2021). Some psychiatric disorders, such as schizophrenia, anxiety, and depression, show an onset of symptoms towards the end of this maturational period (Paus et al., 2008).
Plastic changes induced by lifestyle in the young and immature brain, are useful to cope with normal and pathological age-related cognitive decline (Gelfo et al., 2018; O’Leary et al., 2018), likely, through establishment and implementation of the so-called “brain reserve” (Kempermann, 2008; Walhovd et al., 2014; Zolochevska and Taglialatela, 2016; McQuail et al., 2021). It is now clear that beneficial lifestyle choices may promote changes that enhance cognition, such as angiogenesis, synaptogenesis, and neurogenesis (Lindenberger, 2014). According to a recent interpretation, critical periods, in which experience instructs neural networks to develop into a configuration that cannot be replaced by alternative connectivity patterns, are different from sensitive periods. Experience leads to many possible network configurations or connectivity patterns that can compensate for each other and are subjected to remodeling (Dehorter and Del Pino, 2020). Advances regarding the use of developmental manipulations in the immature brain of animal models are the
basic preclinical systems that will allow the future translatability of timely interventions into clinical applications for neurodevelopmental disorders such as intellectual disability, autism spectrum disorders, and schizophrenia. Nevertheless, besides the involvement of neurotrophins, semaphorins, and GABA deficits (Tapia-Arancibia et al., 2008; Gelfo et al., 2018; Chini and Hanganu-Opatz, 2021; Gao et al., 2021), little is known about the molecular substrates regulating the relationship between plasticity, maturation, and neurological disorders. Since different sensitivity in plastic responses can be found at different, critical ages (postnatal stages, adolescence, youth, aging), it is not easy to translate results obtained in animal models (especially short-living laboratory rodents) to the lifespan and lifestyle of long-lived species such as humans. In summary, the evident involvement of plasticity and brain maturation across a wide range of psychiatric and neurological disorders (both in their pathogenesis and in the perspective of their prevention/treatment) must find more solid neurobiological bases, as well as further knowledge in the translating time science to enhance interpretation of findings from animal models-based studies (Clancy et al., 2001; Charvet, 2021).”
- We inserted 15 new references in addressing points (1) and (3).
References for point 1 are:
Selemon, L.D.; Zecevic, N. Schizophrenia: a tale of two critical periods for prefrontal cortical
development. Transl. Psychiatry 2015, 5, e623.
De Berardis, D.; De Filippis, S.; Masi, G.; Vicari, S.; Zuddas, A. A neurodevelopment approach for
a transitional model of early onset schizophrenia. Brain Sci. 2021, 11, 275.
Gao, W-J.; Yang, S-S.; Mack, N.R.; Chamberlin, L.A. Aberrant maturation and connectivity of
prefrontal cortex in schizophrenia contribution of NMDA receptor development and hypofunction. Mol. Psychiatry 2021, Epub.
Dehorter, N.; Del Pino, I. Shifting developmental trajectories during critical periods of brain
formation. Front. Cell. Neurosci. 2020, 14, 283.
Schor, N.F.; Bianchi, D.W. Neurodevelopmental clues to neurodegeneration. Pediatr. Neurol. 2021,
123, 67-76.
Walhovd, K.B.; Tamnes, C.K.; Fjell, A.M. Brain structural maturation and the foundations of
cognitive behavioral development. Curr. Opin. Neurol. 2014, 27, 176-184.
Morton, P.D.; Ishibashi, N.; Jonas, R.A. Neurodevelopmental abnormalities and congenital heart
disease insights into altered brain maturation. Circ. Res. 2017, 120(6), 960-977.
Chini, M.; Hanganu-Opatz, I.L. Prefrontal cortex development in health and disease: Lessons from
rodents and humans. Trends Neurosci. 2021, 44(3), 227-240.
Paus, T.; Keshavan, M.; Giedd, J.N. Why do many psychiatric disorders emerge during adolescence?
Nat. Rev. Neurosci. 2008, 9(12), 947-957.
Lindenberger, U. Human cognitive aging: Corriger la fortune? Science 2014, 346(6209), 572-578.
Fatemi, S.H.; Folsom, T.D. The neurodevelopmental hypothesis of schizophrenia, revisited.
Schizophr. Bull. 2009, 35, 528-548.
O'Leary, J.D.; Hoban, A.E.; Murphy, A.; O'Leary, O.F.; Cryan, J.F.; Nolan, Y.M. Differential effects
of adolescent and adult-initiated exercise on cognition and hippocampal neurogenesis. Hippocampus 2019, 29, 352-365.
Kaufmann, T.; van der Meer, D.; Doan, N.T., Schwarz, E.; Lund, M.J., Agartz I.; et al. Common ain disorders are associated with heritable patterns of apparent aging of the brain. Nat. Neurosci. 019, 22(10):1617-1623.
References for point 2 are:
Gelfo, F.; Mandolesi, L.; Serra, L.; Sorrentino, G.; Caltagirone, C. The neuroprotective effects of
experience on cognitive functions: Evidence from animal studies on the neurobiological bases of brain reserve. Neuroscience 2018, 370, 218-235.
Tapia-Arancibia, L.; Aliagad, E.; Silhol, M.; Arancibia, S. New insights into brain BDNF function in
normal aging and Alzheimer disease. Brain Res. Rev. 2008, 59(1), 201-220.

Reviewer 2 Report

This is an excellent review on brain plasticity. It covers several important views on the plasticity phenomenon, and contains interesting original ideas on the molecular mechanisms of neurogenesis as well as some approaches that might be rather useful in the critical analysis of experimental data obtained with different (on species, origin, age, etc.) models of brain plasticity.

Author Response

We thank the Reviewer for appreciating our work